# Quad-Band Metamaterial Perfect Absorber with High Shielding Effectiveness Using Double X-Shaped Ring Resonator

**DOI:** 10.3390/ma16124405

**Published:** 2023-06-15

**Authors:** MST Ishrat Jahan, Mohammad Rashed Iqbal Faruque, Md Bellal Hossain, Mayeen Uddin Khandaker, Fahmi Elsayed, Mohammad Salman, Hamid Osman

**Affiliations:** 1Space Science Centre (ANGKASA), Institute of Climate Change (IPI), Universiti Kebangsaan Malaysia, Bangi 43600, Malaysia; 2Centre for Applied Physics and Radiation Technologies, School of Engineering and Technology, Sunway University, Bandar Sunway, Petaling Jaya 47500, Malaysia; 3Department of General Educational Development, Faculty of Science and Information Technology, Daffodil International University, DIU Rd, Dhaka 1341, Bangladesh; 4College of Engineering & Technology, American University of the Middle East, Egaila 54200, Kuwait; 5Department of Radiological Sciences, College of Applied Medical Sciences, Taif University, Taif 21944, Saudi Arabia

**Keywords:** MPA, sensing, pressure, moisture, density

## Abstract

This study assesses quad-band metamaterial perfect absorbers (MPAs) based on a double X-shaped ring resonator for electromagnetic interference (EMI) shielding applications. EMI shielding applications are primarily concerned with the shielding effectiveness values where the resonance is uniformly or non-sequentially modulated depending on the reflection and absorption behaviour. The proposed unit cell consists of double X-shaped ring resonators, a dielectric substrate of Rogers RT5870 with 1.575 mm thickness, a sensing layer, and a copper ground layer. The presented MPA yielded maximum absorptions of 99.9%, 99.9%, 99.9%, and 99.8% at 4.87 GHz, 7.49 GHz, 11.78 GHz, and 13.09 GHz resonance frequencies for the transverse electric (TE) and transverse magnetic (TM) modes at a normal polarisation angle. When the electromagnetic (EM) field with the surface current flow was investigated, the mechanisms of quad-band perfect absorption were revealed. Moreover, the theoretical analysis indicated that the MPA provides a shielding effectiveness of more than 45 dB across all bands in both TE and TM modes. An analogous circuit demonstrated that it could yield superior MPAs using the ADS software. Based on the findings, the suggested MPA is anticipated to be valuable for EMI shielding purposes.

## 1. Introduction

Metamaterial (MM) is a term used to describe a material with an unnatural architecture and exceptional performance [1,2]. In the past few decades, metamaterial absorbers (MAs) have gained popularity for their outstanding absorbing qualities, including their ultra-thin thickness, compact size, minimal cost system, and readily controllable characteristics [3]. Due to their unique electromagnetic (EM) characteristics, MMs may be used widely [4]. The evolution of invisible cloaking [5], lensing [6], perfect absorbers [7], wireless antennas [8], filters [9], energy harvesting [10], sensing [11], radar and satellite communications [12], and seismic detection are just a few of the areas that are motivated by the MMs’ fundamental qualities. Square-ringed resonators are often used in metamaterial architecture because of their simple format and flexibility [13]. However, MM-based microwave sensing has multiple advantages and disadvantages. While certain sensors, such as those used in microfluidic sensing applications, have a high-quality factor, others may be cheap yet have high sensitivity [14,15].

Terahertz sensors which contained a square split-ring resonating with four gaps and a recursive square-ring were developed. When the permittivity of the MM structure changes, the two frequency components form corresponding frequencies on the transmitting spectra curves between 0.1 and 1.9 THz. Each resonant trough exhibited different refractive indices [16]. The proposed MA-based sensor analysis demonstrated the formation of pressure, temperature, density, and humidity sensing [17]. The authors of [17], after achieving perfect absorbance at 6.46 GHz and 7.68 GHz, and an overall absorption rate of 97% per unit of 35 mm × 35 mm MA-based sensing construction, had a cell of squares and resonators. Bakir et al. presented MA with an absorption value of 91.8% at a resonance of 4.2 GHz using an RR-4 substrate (30 mm length × 24 mm width unit cell) [18]. Another study [19] compared the EM interaction response of a hybrid gold–silicon cell with a novel asymmetric design to that of an all-silicon equivalent. However, the gold-based configuration is costly and requires complex maintenance. The authors of [13] made a design to manufacture a unique MM-based sensor to identify substances in the 8 to 12 GHz frequency spectrum, with the resonance shifting by 250 MHz, 200 MHz, 250 MHz, 150 MHz, and 50 MHz for the specimens. The authors of [20] proposed a novel MA-based sensor to identify liquid materials based on their electrical characteristics. Gammadion waveguides employed a copper metallic layer on top of an FR4 substrate with a 2 mm air gap between the copper grounded plane and the resonator’s back to change the resonance responses. Split-ring resonator-based microwave sensors could detect various fluid specimen quantities using detailed permittivity characterisation [20]. An asymmetrical semi-circular split-ring resonance was developed to detect glucose concentrations in the microliter volume [21]. A customised split-ring resonator was designed to detect an electric field, by localising high EM. In another study, SRR resonator-based sensors were constructed using concentric circle sets of bands for industrial, scientific, and medical applications [22]. Meanwhile, ref. [23] presented a susceptible class of meta surface-based nitrogen dioxide (NO_2_) gas sensors that operate in the telecom C band at the 1550 nm line, displaying considerable fluctuations in the reflection coefficient following NO_2_ molecule absorption. The authors used a polymer-based (polyvinylidene fluoride—PVDF, or polyimide—PI) motif consisting of half-rings, rods, or discs with specific sizes and orientations placed on a gold substrate. This setup is very costly. Hossain et al. [8] presented a triple band absorption at frequencies of 5.376, 10.32, and 12.25 GHz with near unity absorption, specifically designed for EMI shielding purposes. The absorber demonstrates exceptional shielding performance, surpassing 40 dB for TE mode at each absorption frequency. A PMA utilising a double elliptical resonator achieved a quadruple band absorption, operating at frequencies of 3.46 GHz, 6.44 GHz, 7.89 GHz, and 12.44 GHz, with absorption rates exceeding 97% [24]. This PMA design was specifically developed for EMI shielding applications. The absorber exhibits shielding capabilities, with a TE mode surpassing 35 dB at each absorption frequency. Nevertheless, there is a deficiency of research focused on developing quad-band or multiband absorbers with higher shielding effectiveness.

This study investigated a quad-band metamaterial perfect absorber (MPA) based on a double X-shaped ring resonator for EMI shielding applications. The proposed design of the double X-shaped resonator consists of four layers: a metallic patch, a dielectric substrate made using Rogers RT5870 with 1.575 mm thickness, a sensing layer, and a copper ground layer with 0.035 mm thickness. The proposed MPA with a quad-band for EMI shielding applications using four resonance frequencies covered C, X, and Ku bands. The cell size of the suggested MPA unit was 8 mm × 8 mm. This cell size yielded excellent absorbance of over 99.9%, 99.9%, 99.9%, and 99.8% over the resonance frequency ranges of 4.87, 7.49, 11.78, and 13.09 GHz. The unit cell also possessed full width at half maximum values of 0.2 GHz, 0.8 GHz, 0.5 GHz, and 0.3 GHz with quality factors of 24.35, 9.28, 23.56, and 43.64 for both TM and TE modes, respectively. To understand the resonance phenomenon, the surface current, electric, and magnetic field fluctuations were investigated. The novelty of the proposed work is that the double X-shaped square split ring resonator-based MPA is of a unique design. The absorption was exceedingly strong under normal polarisation conditions in a range of frequencies (4.87, 7.49, 11.78, and 13.09 GHz), obtaining 100% absorption. The MPA also demonstrates a shielding effectiveness of 68 dB, 87 dB, 83 dB, and 63 dB at absorption peaks of 4.87, 7.49, 11.78, and 13.09 GHz, respectively, for the TE mode; 64 dB, 101 dB, 79 dB, and 46 dB at absorption peaks of 4.87, 7.49, 11.78, and 13.09 GHz, respectively, for the TM mode, indicating the electromagnetic wave’s strength prior to and following the shielding process. The better performance of the proposed MPA is suitable for a wide range of applications, such as EMI shielding, multiple sensing, and C, X, and Ku band application. Moreover, the proposed MPA provided high Q factor and relative absorbance bandwidth (RAB) for the two modes while achieving perfect absorbance performance. The proposed MPA configuration was thoroughly investigated and parametrically analysed for better performance. Finally, the equivalent circuit and ADS outcomes demonstrated an outstanding performance of the recommended MPA.

## 2. Design and Method of the Proposed Unit Cell

Figure 1 demonstrates the projected MPA. The projected MPA unit cell is 8 mm × 8 mm long and wide. The proposed design comprises a square metallic patch with a Rogers (lossy) substrate to achieve insensitive polarisation features and quad-unique resonance frequencies. The front of the unit cell is depicted in Figure 1a and is marked with symbols. The suggested MPA in this study consists of four layers: the plate of pebbles that works as an upper edge is called the unit cell layer, a substrate surrounded by a dielectric material, a sensing layer, and a metal ground surface on the bottom layer. Copper metals with a thickness of 0.035 mm and electric conductivity of σ = 5.8 × 10^7^ S/m were used to design the metallic resonators and ground layer. The one factor preventing absorption in this scenario is that the copper sheet used was substantially thicker than the type of surface region in the EM frequencies. Figure 1c depicts the bottom viewpoint. The parameters of the sensor layers are dielectric constant (fixed at 10), loss tangent (0.002), and the thickness of the layer (0.2 mm). The sensor layer was projected behind the ground and substrate layer. Figure 1b presents the side view of the proposed unit cell for better understanding. Besides that, the Rogers RT 5870 dielectric substrate was selected as an insulating material due to its key benefits, namely minimum loss material. The following elements are used in the dielectric substrate: thickness, h = 1.575 mm; loss tangent, tan δ = 0.0012; permeability, µ = 1; and permittivity, ε_r_ = 2.33. As a result, superior mechanical strength, consistent electrical qualities across a broad frequency range, and a well-established substance comprising two types of glass clothes produced a relatively reasonably priced substrate. The specifications of the resonator employed in the present analysis are displayed in Table 1.

Numerical models were run in the CST Microwave Studio^®^ 2019 to highlight the features of the proposed configuration based on the full-wave finite integration technique (FIT) for the frequency-domain solver. As illustrated in Figure 1d, the suggested unit cell was considered an infinite lattice structure executed using the unit cell boundary conditions in both the x and y directions, allowing an open z direction for the incidence electromagnetic waves to travel down the *Z*-axis. The x and y polarisation Floquet connectors act as transmitters or detectors of EM incident waves. The distance between the component and the microwave sources was 4.75 mm. Moreover, the necessary calculations were performed using the frequency domain solver. Accurate mathematical results were yielded for the influence of polarisation on EM propagating in the z direction.

The absorbance value for frequencies is described as *A*(*ω*) = 1 − *R*(*ω*) − *T*(*ω*), where *A* (*ω*), *R* (*ω*), and *T* (*ω*) indicate the system’s absorption, reflection, and transmission, accordingly [25]. At a particular frequency range, the absorbance *A* (*ω*) can be optimised by minimising the reflection *R* (*ω*) = *|S*11*|*^2^ and transmission *T* (*ω*) = *|S*21*|*^2^ [26]. Absorbance is commonly described as *A*(*ω*) = 1 − *R*(*ω*) due to the presence of the copper ground layer. Moreover, transmissions were nil throughout the entire domain, *T*(*ω*) = 0 [27]. It is reasonable for differently sized resonators to produce absorbance at specific resonances. The combined effects of these peaks created optimal absorbance throughout several resonance frequency regions [28]. According to the literature, various strategies have been proposed to raise the quality factor of these sensors, such as printing devices using a smaller loss of dielectric substrates [14,29]. It is feasible to adjust for a high-quality factor based on absorptions [30]. The formula *Q* = *f*/*FWHM* can be used to check the quality factor using resonance frequency (*f*) and *FWHM* meaning full width at half maximum [31]. For calculation purposes, *f_max_* and *f_min_* denoting higher and lower frequencies equivalent to the absorbance peak with an absorption intensity exceeding 90% were used to establish the *RAB* to examine the absorbance performance of the MPA [32]. The formula is as follows: *RAB =* 2 $\frac{f_{max-}f_{min}}{f_{max+}f_{min}}$ × 100%.

Shielding effectiveness (*SE*) is a crucial metric for measuring the effectiveness of a designed MPA structure in blocking electromagnetic waves at a specific frequency. It serves as a key factor in evaluating the performance of EMI shielding. The *SE* value, which determines how well the shielding operates, is calculated using a formula derived from [8]. This calculation incorporates the reflection and absorption coefficients and is expressed on a logarithmic scale.
(1)$$SE\left(dB\right)={SE}_{R}+{SE}_{A}$$
(2)$$SE\left(dB\right)=10\log\left(\frac{1}{1-{S_{11}}^{2}}\right)+10\log\left(\frac{1-{S_{11}}^{2}}{{S_{12}}^{2}}\right)$$

## 3. Results and Discussions

### 3.1. Absorptions

The MPA reflection, transmission, and absorption characteristics of the TE and TM modes are illustrated in Figure 2a,b. The proposed model displayed quad-bands with an absorption coefficient of 99.9%, 99.9%, 99.9%, and 99.8% for resonance frequencies at 4.87 GHz, 7.49 GHz, 11.78 GHz, and 13.09 GHz for the TE and TM modes. The FWHM values for both modes at all frequencies were 0.2, 0.8, 0.5, and 0.3 GHz, with quality factors of 24.35, 9.28, 23.56, and 43.64, respectively.

The proposed structure exhibited a high Q factor suitable for sensing capability. The quality factor, frequently described as the frequencies divided by the FWHM values, is an essential indication when considering whether to use the resonance modes. The measured quality factor value must be higher if the sensing capability can be increased. The RAB of MPA went up to 6.2%, 12.8%, 9.4%, and 2.4% for the two modes, demonstrating the strong absorbance capabilities of the proposed design. Table 2 provides an overview of the FWHM, Q factor, and RAB for each frequency response for the two modes. The RAB was derived using the RAB equation to evaluate the absorption capability of the MM.

### 3.2. The Expected Development of MPAs

Figure 3 depicts the optimisation method for the proposed unit cell design. There are four different parts to the proposed unit cell MPA design. Varied arrangements of the copper patch in varying ways altered the electrical and magnetic resonance in which the resonances were moved due to the reciprocal capacitance impact of the MPA unit cell. The absorbance displays of the multiple stages involved black, violet, green, and red curves. The first configuration with a square-shaped split ring and two split gap resonators produced 99%, 99%, and 98% absorbance with resonance frequencies at 4.87, 7.50, and 13.02 GHz. Meanwhile, the structure with a split ring and a single X-shaped resonator with two split gaps yielded absorbances of 99%, 99%, and 98% at 4.87 GHz, 7.50 GHz, and 12.35 GHz frequencies. The modified third design with an X-shaped resonator and one split gap exhibited a triple band absorption with an overall 99% absorbance at frequencies of 4.87, 7.50, and 12.62 GHz. The final design combined the double X-shaped resonator, resulting in the proposed unit cell. This combined configuration yielded an ideal quad-band absorbance.

The copper ground plate was inserted beneath the dielectric material to create an absorbance arrangement without a sensor layer. Increased sensitivities were achieved by placing one ring resonator before the planned MPA configuration. Adding the double X-shaped resonator design boosted the absorption and adjusted the absorbance frequencies to the location and specifications of the split. Resonators were positioned on the front edge, with a dielectric substrate as support.

### 3.3. Investigating Maximum Absorption for Parametric Survey

The process was characterised, and the desired absorption was assured by evaluating the variations in the physical measuring elements. Each element can be described as a variable for monitoring performance at a certain point in time, while another remains unchanged. Accordingly, the structure has been divided into several portions denoted as l, r, and s, as illustrated in Figure 1a, whereby each component’s effects were assessed through numerical computations.

#### 3.3.1. Changing the Width of the Square Ring

The width of the ring was modified from 1 mm to 1.8 mm for the TE and TM modes. According to Figure 4, the first, second, and third cases yielded 99% of the overall absorption (heights of 1 mm, 1.2 mm, and 1.4 mm) for all peaks in the two modes while the frequency moved slightly to the left by approximately 0.1 GHz. However, the frequencies shifted over 0.3 GHz for the TE mode as in the fifth case (height = 1.8 mm), reaching 7.49 GHz and 11.78 GHz, whereas 4.87, 7.49, and 13.09 GHz were achieved for the TM mode, making them less effective. In addition, the frequencies of 7.49 and 11.78 GHz for both modes shifted right by about 0.2 GHz for the sixth case (height = 2 mm). Nevertheless, perfect absorbance was achieved for the fourth case (height = 1.6 mm), where the absorption rate was greater than 99% in both the modes.

#### 3.3.2. Modifying the Thickness of the Outer X Resonator

The thickness of the outer X resonator was modified between 0.2 and 0.7 mm in this study. Figure 5a,b depict the intended result for TE and TM modes. The diagram shows whether all peaks have been shifted left by around 0.2 GHz. The absorption rates for all cases were approximately 99%. For cases five and six (thickness 0.6 and 0.7 mm), triple resonance was achieved at 4.87 GHz, 7.49 GHz, and 11.78 GHz, with absorption values of 99.5%, 99.9%, and 99.9%, although the frequency might be shifted to the left by 0.2 GHz. Comparatively, case one (thickness 0.2 mm) yielded 99.9% absorption for both modes, indicating that MPA was attained.

#### 3.3.3. Changing the Thickness of the Inner X Resonator

The thickness of the internal X resonator varied from 0.2 mm to 0.7 mm. Figure 6a,b illustrate the outcomes for both modes. Figure 6 demonstrates that shifting all peaks by 0.1 GHz can induce the frequency shift to the left. The first case (thickness = 0.2 mm) exhibited maximum absorption as the absorbance rate exceeded 99%.

#### 3.3.4. Evaluating Different Substrates

Various substrates were used to evaluate the efficiency of the proposed unit cell MPA. Figure 7a,b present the absorption performance for various substrates for both modes. Five different materials were used as the substrate layers for modelling and were evaluated for their performances in both modes: Rogers RO3003 (lossy) with a thickness of 1.52 mm, FR-4 (lossy) with 1.6 mm thickness, Rogers RT 5880 (lossy) with 1.57 mm thickness, Rogers RT 6002 (lossy) with 1.524 mm thickness, and Rogers RT 5870 (lossy) with 1.575 mm thickness (proposed). The frequencies recorded were 4.67 GHz, 7.08 GHz, 10.48 GHz, and 12.03 GHz for Rogers RO3003 (lossy) and Rogers RT 6002 (lossy), with an infant absorption peak at 13.04 GHz at an absorption of 70% for both modes. A dual resonance was achieved at 6.02 GHz and 9.12 GHz with 98% absorption for the FR-4 substrate. At a frequency of 9.12 GHz, two infant absorption peaks were observed with 82% and 30% absorption rates for both the modes. Meanwhile, Rogers RT5880 (lossy) substrate frequencies were 5.08 GHz, 7.88 GHz, 12.10 GHz, and 13.75 GHz. Comparatively, RT5870 (lossy) substrate yielded perfect absorbance with an absorption rate of more than 99% for the TE and TM modes.

#### 3.3.5. Absorption Behaviour Analysis as a Pressure Sensor

It is assumed that the proposed sensor layer is filled with air and that its thickness changes depending on the applied pressure [33] for TE and TM modes, as depicted in Figure 8a,b. The evaluation was performed numerically by increasing the thickness of the sensor layer, leading to a change in pressure. Five distinct sensor layer thicknesses were assessed in this study, 0.2 mm, 0.4 mm, 0.6 mm, 0.8 mm, and 1 mm, to demonstrate the absorption performance (Figure 8). Five different air cushion widths were particularly chosen for the sensor layer. Due to the inverse relationship between the capacitor and plate spacing, the capacitor value dropped, and the resonance increased (1/√LC) when the distance widened. Since air is an isolating substance, its thickness impacts the absorption [*A*(*w*)]. *A*(*w*) is the key criterion for all sensor applications in this study. Absorption remained at more than 99% for the frequency components of 4.87, 7.49, 11.78, and 13.09 GHz at 0.2 mm, 0.4 mm, 0.6 mm, 0.8 mm, and 1 mm thickness for the sensor layer (Figure 8). Based on the observation, absorption levels were better when the sensor layer thicknesses were 0.2 mm and 0.3 mm. Based on Figure 8, a 0.2 mm change in sensor layer thickness resulted in a 0.2 GHz shift in resonance frequency, indicating that the sensor application based on this absorber is susceptible to sensor layer thickness variations. Moreover, pressure can directly influence system resonance due to the capacitive effect, which is proportional to the thickness of the air gap. Many automotive, medical, industrial, consumer, and construction systems depend on precise and steady pressure readings for reliable functionality. Hence, MPA-based pressure sensors are employed in these types of applications.

#### 3.3.6. Absorption Behaviour Analysis as a Moisture Sensor

In this application, it is expected that the sensor layer is covered in sandy soil and the soil represents the moisture content for which the resonance frequencies were recorded [33]. Figure 9a,b indicated that both modes’ absorbance performance can vary based on volumetric moisture. For the respective volumetric moistures, sensor layer thicknesses of 0.2 mm, 0.6 mm, 1 mm, and 1.4 mm were taken for resonances at 4.87 GHz, 7.49 GHz, 11.78 GHz, and 13.09 GHz. The resonance frequencies shifted to the left as the moisture content increased. The absorbance rate was more than 99% when changed based on the sandy soil’s loss tangent and permittivity coefficients. As intended, the volumetric moisture and frequencies of approximately 0.4 GHz exhibited linear shifting. Generally, the moisture content in agriculture and textile products is an important aspect affecting the shelf life of the items. Sandy soil moisture content with absorption frequency was modelled for this purpose.

#### 3.3.7. Absorption Behaviour Analysis as a Density Sensor

Every material has specific density parameters that vary depending on the environment in which it is utilised. Density sensing is a significant application assessed in this study. The sensor layer can be used to detect the density of a material if constructed appropriately. In this study, the sensor layer was assumed to be filled with different materials of varying densities [33]. Figure 10a,b display the expected outcome for the TE and TM modes. The sensing layer thickness was maintained at 0.2 mm. The sensor layer consists of four different materials: Arlon AD260, Arlon AD320, Arlon AD430, and Arlon AD600. The resonance frequencies recorded for the four materials (AD260, AD320, AD430, and AD600) were 4.87 GHz, 7.49 GHz, 11.78 GHz, and 13.09 GHz, respectively. The maximum absorbance rate of approximately 99% was achieved when AD600-type material was used as a sensor layer. The dielectric properties of AD260, AD320, AD430, and AD600 were 2.6, 3.2, 4.3, and 6, respectively. When the electrical conductivity of the materials employed in the sensor layer increased, the absorption coefficient sensor frequency shifted to the right. The overall variation determined was 0.1 GHz, indicating that the absorbance was sensitive to changes in the electrical conductivity of the sensor layer. This connection serves as the foundation for all sensor applications in this architecture. However, due to the material damage in Arlon 350, Arlon 400, and Arlon 450 together with other manufacturing mistakes, the amplitude of the signals differed by 5% since *ε* = *ε*′ − *j*
*ε*″. In short, the permittivity of the material changes over time due to environmental features or chemicals/biochemical materials, as they exhibit varied permittivity characteristics [34].

### 3.4. Stabilisation of Polarisation and Incident Angles

The proposed MPA must reliably absorb incident electromagnetic radiation throughout all oblique incident and polarisation angles for real-world applications. For this purpose, the theoretical investigation was conducted for the proposed MPA for different oblique incident and polarisation angles. The proposed MPA’s estimated absorbances for the TE and TM modes are depicted in Figure 11a,b, with various incident angles. Meanwhile, Figure 11c,d depict the expected MPA absorbance with different polarisation angles for both modes. The proposed MPAs exhibited quad-band absorptions at 4.87 GHz, 7.49 GHz, 11.78 GHz, and 13.09 GHz, with absorbances of 99.9%, 99.9%, 99.9%, and 99.8% when φ = 0^0^ and θ = 0^0^ for TE and TM modes. At every resonance, the proposed absorber displayed magnetic and electric dipole moments. As shown in Figure 11a,b, the absorber is insensitive to incident angles up to 50^0^ maintaining above 80% absorption, after which point absorption declines as the angle increases. Due to the asymmetrical design of the suggested MPA structure, it is also found that the second and fourth peak absorption gradually decrease as the incident angle increases. According to Figure 11c,d, an absorption figure of more than 60% for both modes is maintained up to a polarisation angle of 20^0^ but is somewhat moved to the right at higher angles. These undesirable changes are most likely the result of cross coupling between the components of the resonating patch of the unit cell.

## 4. Surface Current, Magnetic Field, and Electric Field Evaluations

The physical stability of the proposed MPA was evaluated based on surface current, magnetic (H-field), and electric field (E-field) distribution at four absorption frequencies. The surface currents, H-field, and E-field are depicted in Figure 12. According to Figure 12i, magnetism excitation was observed when the surface current passed through the top resonant patched for the four absorption peaks. The maximum current was concentrated at the right-side portion of the square ring resonator at 4.87 GHz (Figure 12a,(i)), two opposite side portions of the square resonator at 7.08 GHz (Figure 12b,(i)), two sides of the outer X resonator at 11.78 GHz (Figure 12c,(i)), and in the external X resonator at 13.09 GHz (Figure 12d,(i)). Remarkably, the H-field and E-field were simultaneously activated at the resonance frequency, causing significant EM light absorption. These surface currents and the H-field in this research show similar values. Moreover, the current surface orientations for all absorption peaks were anti-parallel to the surfaces of the proposed MPA. The magnetic dipole moment produced by these anti-parallel current loops is strongly correlated with the incident EM field. Therefore, the proposed structure enabled the synchronous activation of the magnetic resonances at 4.87 GHz, 7.49 GHz, 11.78 GHz, and 13.09 GHz. Figure 12(ii) depicts the H-field composition for four absorption peak scenarios where surface current and H-field appeared to be in excellent agreement, based on Ampere’s law $\left(\int B.dl=\mu_{0}I\right)$. Meanwhile, E-field is presented in Figure 12(iii). The E-field fluctuation is most significant at the square ring resonator at 4.87 GHz frequency. The two opposed gap corners of the square ring resonator formed an E-field at 7.08 GHz resonance, whereas the double X-shaped resonator achieved a maximum E-field at different frequencies, 11.78 or 13.09 GHz. Based on both field distributions, the electric excitation was caused by the fields condensing along the top resonating region. The electric and magnetic excitation were regulated for the PMA structure to create a powerful absorption property.

## 5. Evaluation of an Equivalent Circuit

The equivalent circuit approach for a proposed MPA unit cell is a mechanism commonly used to evaluate the electrical performance of the absorbency. It is based on the required absorption’s ground surface (copper annealed), which is inserted into the circuit developed using ADS (Advanced Design System software), as shown in Figure 13a. The S_11_ parameter was used as a variable for the resonance frequencies discovered employing simulated results. L1 and L2 were used for bottom copper. C1 is a coupling capacitor connecting the bottom copper with the upper resonator. L3 and L4 inductances represent the square ring, while C4 and C5 capacitance represent the gaps. Thus, L3, C4, L4, and C5 are interconnected. An EM field transverse to the gaps of the component stimulates the capacitor (C) in the circuit. C2 is a coupling capacitor between the square ring and the X-shaped resonator. Inductors L5 and L6 denote the external X-shaped resonator. Meanwhile, C3 is the coupling capacitor between the outer and the inner X-shaped resonators. L3, C4, L4, and C5 represent the inner X-shaped resonator and they are interconnected. The presented unit cell configuration of the circuit design and CST simulations indicated the capability of the proposed design to produce quad resonance frequencies. Figure 13b analyses the S_11_(dB) values from the analogous circuit and the CST program.

## 6. EMI Shielding Applications

The capacity of EMI shielding to block and protect electromagnetic waves is assessed by a crucial factor known as shielding effectiveness (SE). The total SE considers the reflection and absorption coefficients. Figure 14 illustrates the simulated scenario of SE. The PMA’s shielding effectiveness is observed to be 68 dB, 87 dB, 83 dB, and 63 dB at absorption peaks of 4.87, 7.49, 11.78, and 13.09 GHz, respectively, for the TE mode; and 64 dB, 101 dB, 79 dB, and 46 dB at absorption peaks of 4.87, 7.49, 11.78, and 13.09 GHz, respectively, for the TM mode. This indicates that the shielding mechanism predominantly contributes to absorption while having a negligible impact on the reflection behaviour. For most applications, a minimum SE of above 35 dB is required, signifying a 99% attenuation of incident electromagnetic waves.

Table 3 contrasts the suggested MPA results with the previous MA work based on the absorption level, unit cell size, design structure, absorption frequency, and other factors. Besides raising the absorption level, the proposed absorber structure reduced the substrate’s dimensions and thickness. MA, which has been previously described, possessed multi-band properties with better absorption levels due to different types of top resonators and dielectric substrates. Meanwhile, the designed MPA is a more compact and multiple sensor application, revealing quad-band absorptions of 99.9%, 99.9%, 99.9%, and 99.8% at 4.87, 7.49, 11.78, and 13.09 GHz, respectively. This structure also exhibited high Q factors of 24.35, 9.28, 23.56, and 43.64. The estimated RABs were 6.2%, 12.8%, 9.4%, and 2.4% at all absorption frequencies. Table 1 includes references that demonstrate multiple resonance peaks across various frequency bands, exhibiting satisfactory absorption and shielding effectiveness for the TE mode. However, the designed PMA surpasses them by showcasing quad-band resonance and exceptional EMI shielding effectiveness for both TE and TM modes based on simulated outcomes.

## 7. Conclusions

This study presents quad-band MPA based on a double X-shaped ring resonator for EMI shielding applications. Based on the CST Microwave Studio simulation, the unit cell displayed quad-band resonance frequencies at 4.87, 7.49, 11.78, and 13.09 GHz with absorbances of 99.8%, 99.9%, 99.9%, and 99.9%, respectively, together with estimated quality factors of 24.35, 9.28, 23.56, and 43.64 for both modes. The intended unit cell design is unique because it can provide a high Q factor and RABs of 6.2%, 12.8%, 9.4%, and 2.4% for both modes while achieving perfect absorbance performance. It can also provide excellent shielding effectiveness for both TE and TM modes. Based on the surface current, H-field, and E-field distribution analyses, the propagating EM radiation inspects the response and contribution of various unit cell resonance frequencies. Finally, the analogous circuit and ADS results demonstrated the remarkable performance of the proposed MPA. Due to its excellent performance, the presented MPA-based EMI shielding mechanism has the potential to be employed in shielding electronic equipment in future applications.

## Figures and Tables

**Figure 1 materials-16-04405-f001:**
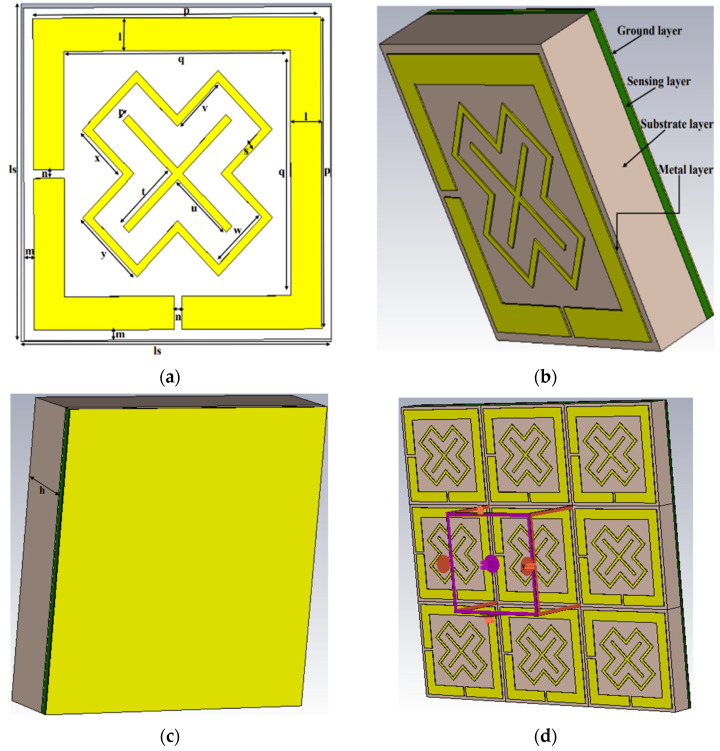
Different views of the proposed MPA (**a**) upper view, (**b**) side view, (**c**) bottom view, and (**d**) simulation geometry.

**Figure 2 materials-16-04405-f002:**
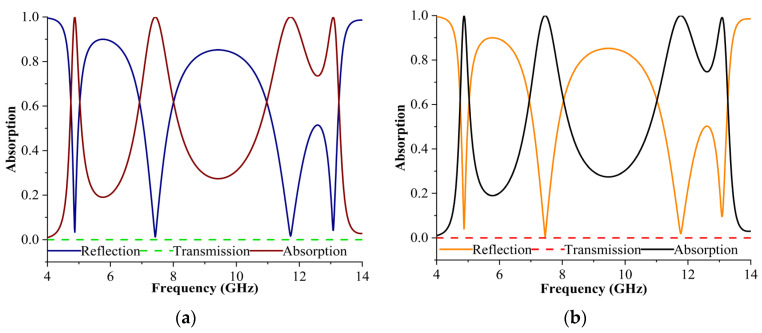
Results of reflection, transmission, and absorption in (**a**) TE (**b**) TM modes.

**Figure 3 materials-16-04405-f003:**
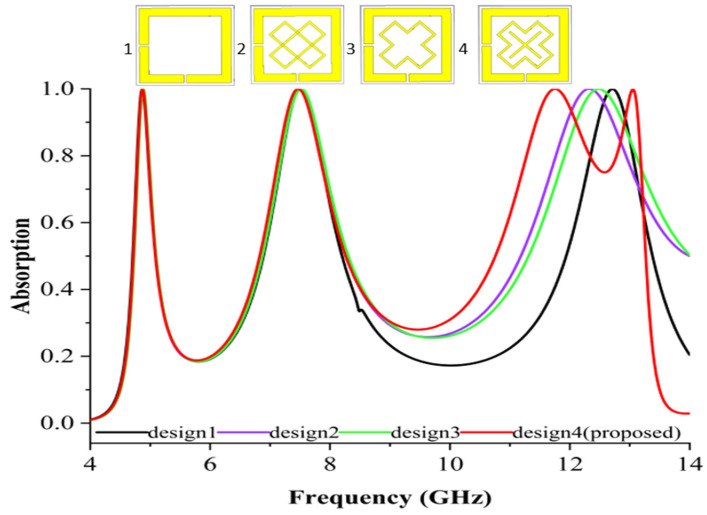
The design optimisation scenario.

**Figure 4 materials-16-04405-f004:**
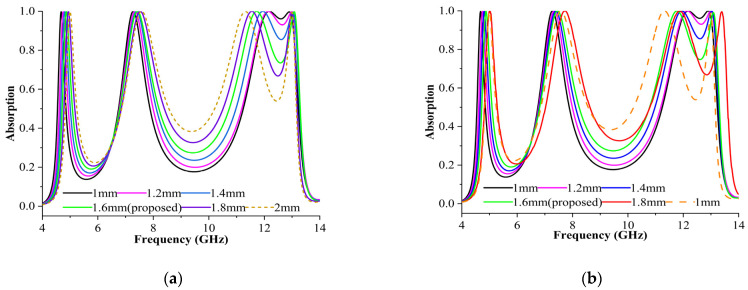
Absorptions for square rings with varying widths in (**a**) TE and (**b**) TM modes.

**Figure 5 materials-16-04405-f005:**
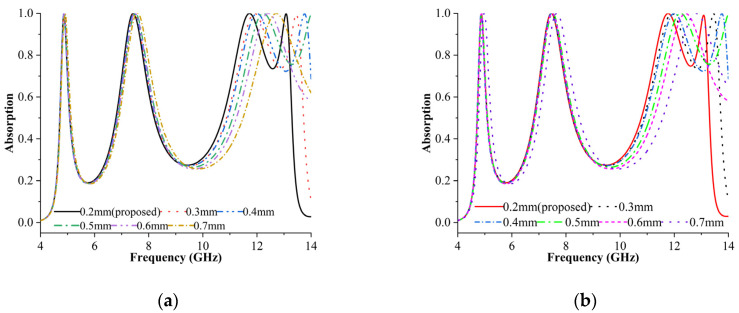
Absorption rates of outer X resonator with varying thicknesses in (**a**) TE and (**b**) TM modes.

**Figure 6 materials-16-04405-f006:**
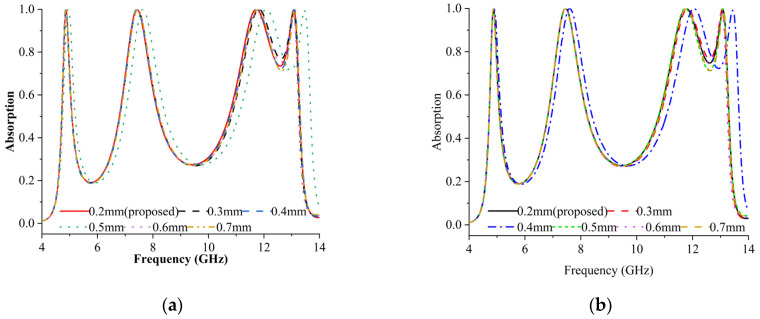
Absorption rates of inner X resonator with varying thicknesses in (**a**) TE and (**b**) TM modes.

**Figure 7 materials-16-04405-f007:**
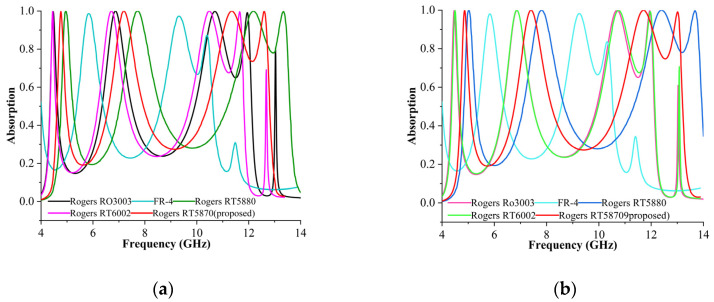
Absorptions for various substrates in (**a**) TE and (**b**) TM modes.

**Figure 8 materials-16-04405-f008:**
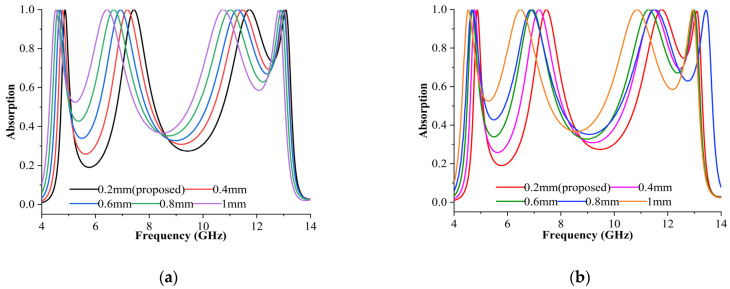
Absorption behaviour analysis as a pressure sensor in (**a**) TE and (**b**) TM modes.

**Figure 9 materials-16-04405-f009:**
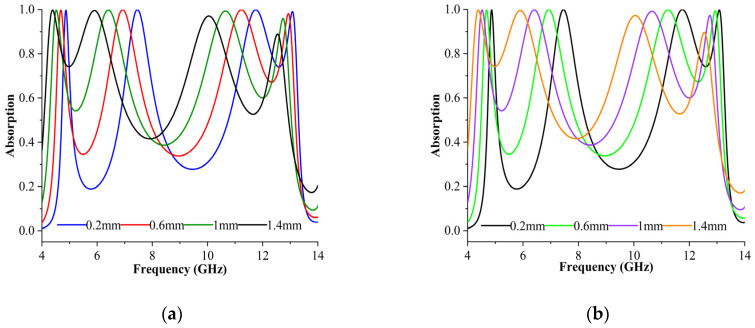
Absorption behaviour analysis as a moisture sensor for (**a**) TE and (**b**) TM modes.

**Figure 10 materials-16-04405-f010:**
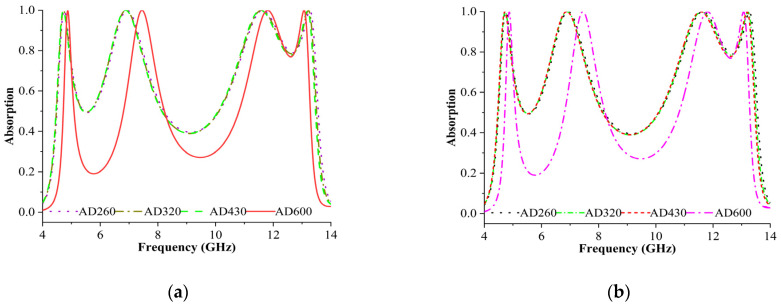
Absorption behaviour analysis as a density sensor in (**a**) TE and (**b**) TM mode.

**Figure 11 materials-16-04405-f011:**
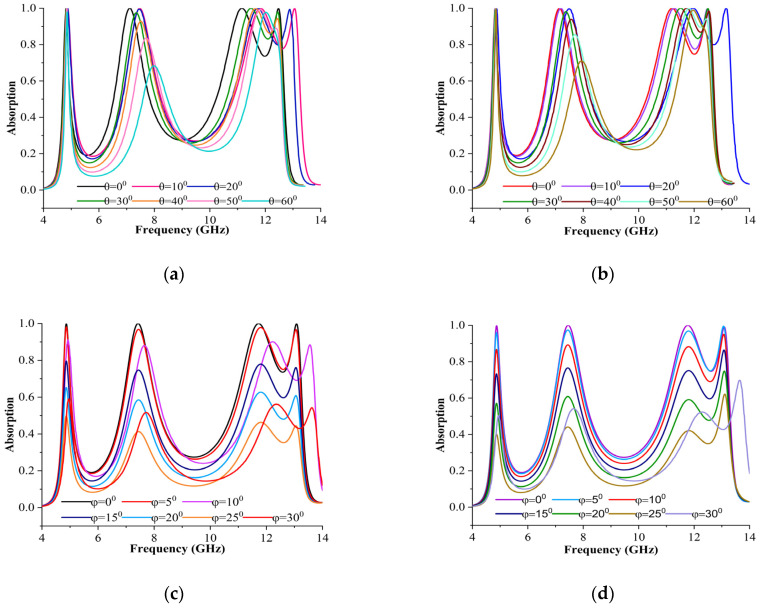
Absorbance performance in (i) different incident angles: (**a**) TE mode, (**b**) TM mode; and (ii) different polarisation angles: (**c**) TE mode, (**d**) TM mode.

**Figure 12 materials-16-04405-f012:**
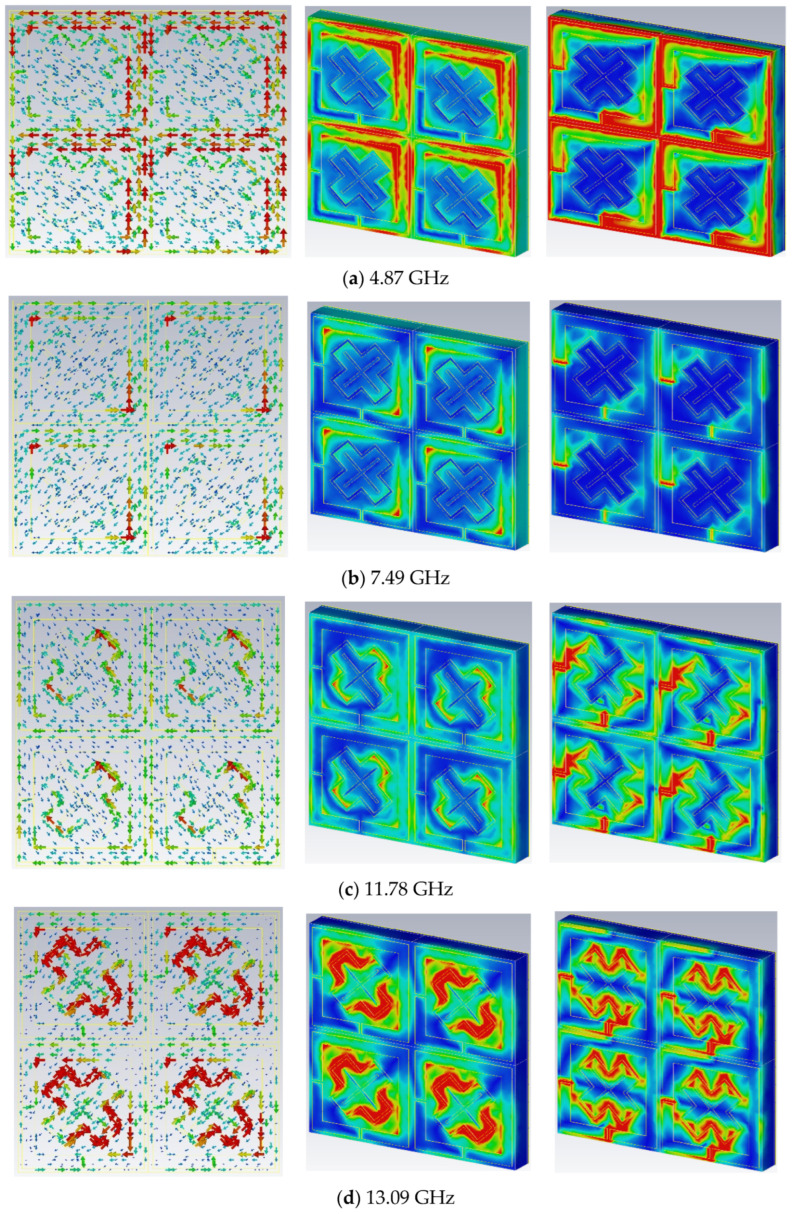
Analyses of surface current, H-field, and E-field at different frequencies: (**a**) 4.87 GHz; (**b**) 7.49 GHz; (**c**) 11.78 GHz; (**d**) 13.09 GHz.

**Figure 13 materials-16-04405-f013:**
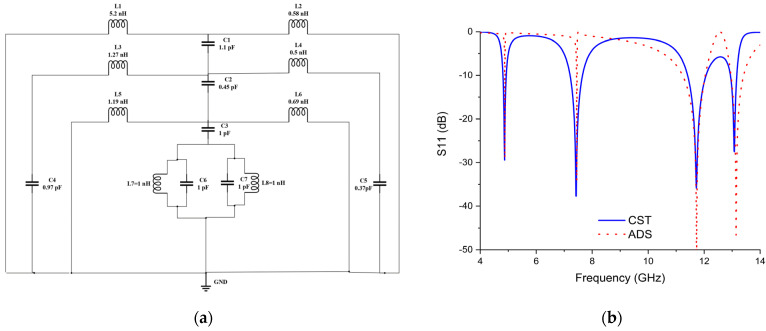
(**a**) Equivalent circuit (**b**) S11 results using CST and ADS simulator.

**Figure 14 materials-16-04405-f014:**
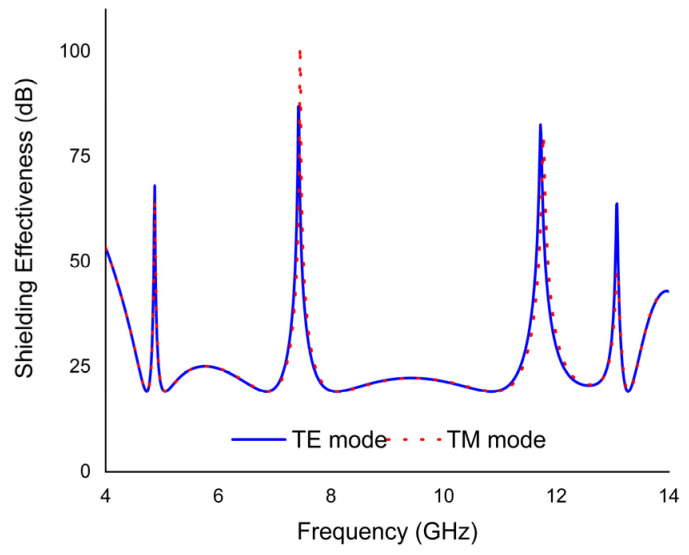
Shielding effectiveness results for TE and TM modes.

**Table 1 materials-16-04405-t001:** The suggested resonator’s design specifications.

Parameters	Value (mm)	Parameters	Value (mm)
ls	8	r	0.2
l	1.6	s	0.2
m	0.5	t	1.8
n	0.2	u	1.8
p	7.5	v	1.5
q	5.9	w	1.6
x	1.5	y	2

**Table 2 materials-16-04405-t002:** The FWHM, Q factor, and RAB data for the TE and TM modes.

Frequency (GHz)	FWHM	Q Factor	*f_max_*; *f_min_*	RAB
4.87	0.2	24.35	9.7; 0.3	6.2%
7.49	0.8	9.36	14.95; 0.95	12.8%
11.78	0.5	23.56	23.5; 1.1	9.4%
13.09	0.3	43.63	25.9; 0.3	2.4%

**Table 3 materials-16-04405-t003:** Comparisons of MMAs.

References	Size (mm)	Frequency Range (GHz)	Design	Absorption Rate (%)	Polarisation Independent	Incident Angle Insensitive	EMI Shielding Application
[7]	43.2	1.34, 2.15, 3.2, 4.31 and 5.41	A cylindrical waveguide	Above 60%	Yes	Yes	No
[8]	10 × 10	5.376, 10.32 and 12.25	Double E -shaped	Above 99%	Yes	Yes	Yes (only TE)
[13]	35 × 35	8 to 12	Oval-shaped wing resonators	Above 83%	No	No	No
[14]	9 × 9	2.21, 4.1, and 5.6	A double split ring enclosed nested meander-line-shaped	Above 90%	No	No	No
[24]	12×12	3.46, 6.44, 7.89 and 12.44	Double Elliptical	Above 97%	Yes	Yes	Yes (only TE)
Proposed design	8 × 8	4.87, 7.49, 11.78, 13.09	double X-shape with a square split ring resonator	Above 99%	-	-	Yes (Both TE and TM), High SE

## Data Availability

All the data are available within the manuscript.

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
