# Peer review of "Quad-Band Metamaterial Perfect Absorber with High Shielding Effectiveness Using Double X-Shaped Ring Resonator"

_materials, 2023, doi:10.3390/ma16124405_

Round 1

Reviewer 1 Report

Major Revision, Extensive English Proofreading. See attached.

Author Response

As  attached.

Reviewer 2 Report

This paper proposes a metasurface configuration with an application in sensors. The concept is good but it is badly written. In particular:

1) The language manipulation is very bad and an extensive review is required. There are various spots where the text is not comprehensible.

2) The authors claim in section 3.3 that the proposed device is insensitive to incidence and polarization angle changes. However, Figure 4 shows a totally different story, especially for the polarization angle change, where a clear degradation is visible even for small changes. This must be changed in Table 2, also.

3) The surface current section is badly written and there are not many conclusions.

4) The parametric analysis must precede the sensor application.

5) There are too many figures that do not provide much information, such as many of the surface current ones, the design from all the angles, etc.

6) The section with sensors is a good option, but the implementation is moderate:

6a) The change in pressure can, indeed, alter the properties of the proposed MPA. How can one measure this change from a practical point of view? Specifically, what device will measure this change and what's the arrangement? Of course, the idea is not to provide full details, but it seems very difficult to make such a measurement in real-life conditions.

6b) What's the connection between the volumetric change of moisture with the electromagnetic parameters? It must be determined in the manuscript.

6c) The density sensor is not defined adequately. The connection between the different materials with the density evaluation must be defined in the manuscript. Moreover, the sensing capabilities are limited.

7) Some additional minor comments:

The paper starts with the acronym MM. It should be first defined

In line 123, the relative permittivity is shown and it must be expressed as er.

In lines 143-144, the square values must be corrected.

Author Response

As attached.

Reviewer 3 Report

In manuscript entitled "Double X-shape with a square split ring resonator based quad band polarization and incident angle insensitive metamaterial perfect absorber for sensing applications", the authors have used double X-shape with a square split ring resonator based quad band polarization for sensing applications. The idea is interesting and the methodology is well-designed. As per my check similarity to the previous literature is 16% which is acceptable. However, there is minor issues which should be considered.

1. The title is not interesting and catchy.

2. The previous studies in this era have not been sited properly and the authors only the recent research while the fondumental studies have been skipped.

3. the authors should highlight the importance of this study and the state of the art.

4. There is lack of comparission between the finding of this study and literature to point out its significance.

5. The conclussion should be more informative and comparitive.

Author Response

As attached.

Round 2

Reviewer 1 Report

The authors have answered all observation. I recommend publication.

Author Response

Thank you for the recommendation for publication.

Reviewer 2 Report

The authors conducted several changes based on the recommendations, but there are still some problems that require further attention:

1) Section 3.4 (Stabilisation of polarisation and incident angles) has many problems. Again, the incident angle stability is in question since beyond 40o the performance of the 2nd resonance (6-8 GHz) degrades considerably for both modes. Additionally, the resonance frequencies 2, 3, and 4 are influenced strongly by different incidence angles based on Figure 8. Consequently, there is no incidence angle insensitivity.

2) The practical implementation for all the sensors must be highlighted explicitly in the paper. In their answer, the authors claim that a transmission line is connected to the microstrip. However, here we do not have any microstrip but a flat surface. Is this correct? If yes, then an appropriate arrangement must be defined.

3) The quality of the figures for section 5 should be improved (especially the color axis).

4) The numbering of the tables is not correct (two Table 3 exist).

Author Response

As attached.

Round 3

Reviewer 2 Report

It seems that the major comments cannot be addressed sufficiently. In particular:

1) Insensitivity is considered to maintain the absorption over 80% for the incident angle and over 60% for the polarization angle. Obviously, this is a totally arbitrary selection. Moreover, insensitivity corresponds to negligible differences. So, the authors must change the insensitivity term both in the analysis and the title of the paper.

2) The changes in the manuscript with respect to the sensors are limited. The authors provide some details in the answer, but the proposed arrangement is not valid since the back side of the metasurface is covered by metal, based on the analysis. Consequently, the back antenna cannot receive any signal. Concluding, the effort to present some sensor applications is interesting, but the implementation is very poor. The whole section should be removed and one paragraph can be written for the possible applications.

If the authors are able to conduct these changes, the suggestion will be positive.

Author Response

As attached.
